

# Insights on early mutational events in SARS-CoV-2 virus reveal founder effects across geographical regions

Carlos Farkas[1,2], Francisco Fuentes-Villalobos[3], Jose Luis Garrido[4], Jody Haigh[1,2] and María Inés Barría[3]

[1] Oncology and Hematology, CancerCare Manitoba, Winnipeg, MB, Canada
[2] Department of Pharmacology and Therapeutics, Rady Faculty of Health Sciences, University of Manitoba, Winnipeg, MB, Canada
[3] Faculty of Biological Sciences, Department of Microbiology, Center of Biotechnology, Universidad de Concepción, Universidad de Concepción, Concepción, Chile
[4] Ichor Biologics LLC, New York, NY, USA

## ABSTRACT

Here we aim to describe early mutational events across samples from publicly available SARS-CoV-2 sequences from the sequence read archive and GenBank repositories. Up until 27 March 2020, we downloaded 50 illumina datasets, mostly from China, USA (WA State) and Australia (VIC). A total of 30 datasets (60%) contain at least a single founder mutation and most of the variants are missense (over 63%). Five-point mutations with clonal (founder) effect were found in USA next-generation sequencing samples. Sequencing samples from North America in GenBank (22 April 2020) present this signature with up to 39% allele frequencies among samples ($n = 1,359$). Australian variant signatures were more diverse than USA samples, but still, clonal events were found in these samples. Mutations in the helicase, encoded by the ORF1ab gene in SARS-CoV-2 were predominant, among others, suggesting that these regions are actively evolving. Finally, we firmly urge that primer sets for diagnosis be carefully designed, since rapidly occurring variants would affect the performance of the reverse transcribed quantitative PCR (RT-qPCR) based viral testing.

Corresponding authors
Carlos Farkas,
carlos.farkaspool@umanitoba.ca
María Inés Barría, mbarriac@udec.cl

## INTRODUCTION

The COVID-19 pandemic caused by a novel 2019 SARS coronavirus, known as SARS-CoV-2, is rapidly spreading worldwide, greatly surpassing the 8,000 total cases of the 2002–2004 SARS coronavirus outbreak (SARS-CoV-1) after 1 month of the initially identified case on 31 December 2019, in Wuhan city, China (*Wilder-Smith, Chiew & Lee, 2020*). As SARS-CoV-2 is human-to-human transmitted, it is a threat to the global population. It is critical to understand SARS-CoV-2 characteristics to deal with this ongoing pandemic and to develop future treatments. SARS-CoV-2 virus is an enveloped, positive-stranded RNA virus with a large genome (29.9 kb) belonging to the family Coronaviridae, order Nidovirales (*De Wit et al., 2016*). One of the striking genomic

features of this novel virus is the presence of a novel furin-like cleavage site in the S-protein of the virus, which differs from SARS-CoV-1 and may have implications for the life cycle and pathogenicity of the novel virus (*Coutard et al., 2020*; *Wu et al., 2020a*). Firstly, it was suggested that SARS-CoV-2 is a close relative of the RaTG13 bat-derived coronavirus (around 88% identity) rather than of SARS-CoV-1 (79% identity) or Middle East respiratory syndrome coronavirus MERS-CoV (50% identity) (*Lu et al., 2020*). Due to this association with bat coronaviruses, it was also argued that SARS-CoV-2 virus has the potential to spread into another species, as bat coronaviruses do (*Hu et al., 2018*). Although bats are likely natural reservoir hosts for SARS-CoV-2, it was recently demonstrated that SARS-CoV-2 is closely related to a pangolin coronavirus (Pangolin-CoV) found in dead Malayan pangolins with a 91.02% identity, the closest relationship found so far for SARS-CoV-2 (*Zhang, Wu & Zhang, 2020*). In that study, genomic analyses revealed that the S1 protein of Pangolin-CoV is related closer to SARS-CoV-2 than to RaTG13 coronavirus. Also, five key amino acid residues involved in the interaction with the human ACE2 receptor are maintained in Pangolin-CoV and SARS-CoV-2, but not in RaTG13 coronavirus. Thus, it is likely pangolins are an intermediate host in the transmission of coronaviruses between bats and humans. In this manner, it was argued SARS-CoV-2 acquired mutations needed for human transmission and will continue to evolve with novel mutations, as the pandemic evolves (*Zhang & Holmes, 2020*). In this scenario, it is expected that diverse signatures of viral variants spread among different populations in the world. Recently, thousands of GenBank sequences from SARS-CoV-19 available at the NCBI virus database were trackable by region, suggesting that the transmission occurred mainly through clonal events due to clustering of the available sequences (*Chen, Allot & Lu, 2020*; *Kupferschmidt, 2020*) (https://www.ncbi.nlm.nih.gov/labs/virus/vssi/#/virus?SeqType_s=Protein&VirusLineage_ss=SARS-CoV-2,%20taxid:2697049). As a proof of concept, in the early beginning of the outbreak in China, sequencing the virus from nine patients from Wuhan in China revealed 99.9% similarity among samples. That finding suggests 2019-nCoV originated from one source within a very short time, supporting clonality of spreading (*Lu et al., 2020*). In this study, we characterized the early mutational events across 50 illumina high-quality datasets publicly available on the sequence read archive repository. A total of 30 out of 50 samples (60%) contained at least a single founder variant and most of the variants across samples are missense (over 63%). SARS-CoV-2 founder variants in WA State, USA (USA-WA) are dissimilar to Australian SARS-CoV-2 founder variants, which were found to be heterogeneous. However, a mutational signature from USA mutations was found in an Australian sample, suggesting a world-wide spread of this molecular signature consisting of five-point variants. Remarkably, mutations in the helicase and ORF1ab proteins of the virus were found more frequently than others, suggesting that these regions continue to actively evolve. As proof of the latter, a single nucleotide polymorphism (SNP) in an Australian sample causes a bona-fide stop codon in the helicase protein. As genetic drift prompts the mutational spectrum of the virus, we recommend frequently sequencing the viral pool in every country to detect the founder events relevant for SARS-CoV-2 testing in each population.

## MATERIALS AND METHODS

### Data collection

Raw illumina sequencing data were downloaded from the following NCBI SRA BioProjects: SRA: PRJNA601736 (Chinese datasets), SRA: PRJNA603194 (Chinese dataset) (*Wu et al., 2020b*), SRA: PRJNA605907 (Chinese datasets) (*Shen et al., 2020*), SRA: PRJNA607948 (USA-WI State datasets), SRA: PRJNA608651 (Nepal dataset), SRA: PRJNA610428 (USA-WA State datasets), SRA: PRJNA612578 (USA-San-Diego dataset), SRA: PRJNA231221 (USA-WA State dataset) (*Sichtig et al., 2019*), SRA: PRJNA613958 (Australian-VIC datasets), SRA: PRJNA231221 (USA-MD State dataset), and SRA: PRJNA614995 (USA-UT datasets). All illumina SRA accessions until 27 March 2020 are depicted in Table S1, sheet 1. Illumina SRA accessions until 22 April 2020 are available in https://github.com/cfarkas/SARS-CoV-2_illumina_analysis and were obtained from SARS-CoV-2 resource at GenBank: https://www.ncbi.nlm.nih.gov/genbank/sars-cov-2-seqs/.

### Data processing

Primer sequences were aligned with bowtie2 aligner (v2.2.6) (*Langmead & Salzberg, 2012*) against SARS-CoV-2 reference genome NC_045512.2, using the following parameters: -D 20 -R 3 -N 0 -L 20 -i S,1,0.50. Illumina Raw reads from whole genome sequencing and amplicon sequencing were trimmed by using fastp tool in default mode (*Chen et al., 2018*) and aligned by using Minimap2 aligner using the preset -ax sr against SARS-CoV-2 reference genome (*Li, 2018*). GenBank fasta sequences form Asia, Europe and North America were also aligned by using minimap2 with the same preset. Samtools v1.9 (using htslib v1.9) (*Li et al., 2009*) was used to sort sam files and index bam files. bcftools v1.9 (part of the samtools framework) was used to obtain depth of coverage in each aligned sample. For variant calling in illumina samples, Strelka2 variant caller was employed in each bam dataset by invoking GermlineWorkflow and outputted variants were filtered by using the "PASS" criteria (*Kim et al., 2018*). Variants from next generation sequencing reads and GenBank alignments were also called with bcftools mpileup with the following parameters: -B -C 50 -d 250. To obtain founder mutations, filtering of called variants was performed with bcftools filter, considering variants only with Mann–Whitney $U$ test of read position bias over 0.1 and the number of high-quality reference alleles divided by high-quality alternate alleles over 0.3. All commands to obtain these computational steps are publicly available at https://github.com/cfarkas/SARS-CoV-2_illumina_analysis.

### SNVs consequences and classification

All SNP and INDELs consequences were assessed in each sample by using snippy haploid variant calling and core genome alignment pipeline: https://github.com/tseemann/snippy. Also, Variant effect annotation tool, employing the variant effect predictor algorithm (VEP) was employed to assess functional effects of variants on SARS-CoV-2 transcripts. (*Hinrichs et al., 2016*; *McLaren et al., 2016*).

## Construction of multiple sequence alignments with GenBank sequences and phylogenetic tree inference

1599 SAR2-CoV-2 GenBank datasets from Asia ($n = 190$), Europe ($n = 40$) and North America ($n = 1,359$) were downloaded on 22 April 2020 from NCBI virus database (https://www.ncbi.nlm.nih.gov/labs/virus/vssi/#/) using as query "Severe acute respiratory syndrome coronavirus 2 (SARS-CoV-2), taxid:2697049". All sequences are publicly available at https://github.com/cfarkas/SARS-CoV-2_illumina_analysis. Merged sequences were also aligned by using MAFFT multiple sequence alignment program version 7.271 (*Katoh & Standley, 2013*) using the—reorder flag. Fasttree version 2.1 was used to infer an approximately-maximum-likelihood phylogenetic tree from the aligned sequences in fasta format by using heuristic neighbor-joining clustering method (*Price, Dehal & Arkin, 2010*). Visualization and editing of the phylogenetic tree were perfomed by using Interactive Tree of Life server (iTOL), collapsing all clades whose average branch length distance was below 0.0002 (*Letunic & Bork, 2019*).

## Primer list obtention

CDC primers currently in use (April 2020) were obtained from https://www.cdc.gov/coronavirus/2019-ncov/lab/rt-pcr-panel-primer-probes.html, generated by the Division of Viral Diseases, National Center for Immunization and Respiratory Diseases, Centers for Disease Control and Prevention, Atlanta, GA, USA. University of Hong-Kong primers currently in use were obtained from https://www.who.int/docs/default-source/coronaviruse/peiris-protocol-16-1-20.pdf?sfvrsn=af1aac73_4 and Institute Pasteur primers currently in use were obtained based on the first sequences of SARS-CoV-2 made available on the GISAID database (*Shu & McCauley, 2017*) on 11 January 2020, available here: https://www.who.int/docs/default-source/coronavirus/real-time-rt-pcr-assays-for-the-detection-of-sars-cov-2-institut-pasteur-paris.pdf?sfvrsn=3662fcb6_2. We also included literature-based primers (*Kim et al., 2020*) and ten primer-BLAST (*Ye et al., 2012*) designed primers against SARS-CoV-2 reference genome NC_045512.2.

# RESULTS

## Inspection of variants reveals well-defined signatures with founder effects across sequenced samples

We aimed to call variants of SARS-CoV-2 datasets sequenced with the Illumina technology, due to its depth and sequencing quality, in terms of error rate (*Nielsen et al., 2011*). As of 27 March 2020 we obtained 282 accession numbers for SARS-CoV-2, from the sequence read archive, containing 27 illumina datasets. By searching in Sequence Read Archive repository (SRA) we added 24 more datasets yielding in total 51 illumina raw datasets to analyse (see Table S1, sheets 1 and 2). From this list, we excluded the Chinese Sample nCoV5 (SRR11059943) due to large gaps in genome coverage, as explained in (*Shen et al., 2020*) and we subsequently worked with these illumina datasets. We aligned each fastq reads against the SARS-CoV-2 reference genome NC_045512.2, corresponding to the initial isolate Wuhan-Hu-1. We checked coverage of each sample by using the Integrative Genomics Viewer tool (*Robinson et al., 2011*) and samtools (Table S1, sheet 2).

Variant calling in each sample by using Strelka2 reveals a diverse number of variants per sample, yielding 137 single nucleotide polymorphisms (SNVs) and nine indels (see Fig. 1A; Table S1, sheet 3). Founder variants were obtained by doing variant calling with bcftools after strict filtering (see Table S1, sheet 2). Remarkably, thirteen out of fourteen datasets from the USA-WA State study (hereafter referred as USA-WA) displayed variants presenting a defined variant signature, consisting in a core of five founder variants at positions 8,782, 17,747, 17,858, 18,060 and 28,144 in the SARS-CoV-2 reference genome, also detected within the 137 SNVs from the next generation sequencing datasets (see Fig. 1B; Table S1, sheets 2 and 3). Mutational landscape analysis of SARS-CoV-2 samples in Australia (Australia-VIC samples, hereafter Australia-VIC) demonstrated that these samples were clearly heterogeneous, displaying a variety of founder mutations per sample but also shared variants were observed within samples. One variant (position 26,144) is present in 5/11 Australia-VIC samples and variants 8,782 and 28,144 from USA-WA signature are also present in 3/11 Australian samples (see Fig. 1C; Table S1, sheet 5). Notably, one Australian-VIC sample (SRR11397717) displayed the same five-point variant signature of USA-WA samples, two samples contain the same variant signature presenting one deletion (SRR11397715 and SRR11397716) and one novel signature (SRR11397728) presents a SNP that creates a stop codon (see Table S1, sheet 5). All of these called variants present mutant allele frequencies near or equal to 100%, evidenced in the number of mutant_alleles/reference_alleles (see mutant allele frequency in Table S1, sheets 4 and 5, respectively) easily visualized in the aligned bam files (see Fig. 1C from USA-WA and Fig. 1D for Australian-VIC samples, respectively). These analyses suggest that these variants were already spread in the infected population in the early days of the outbreak, they are not restricted by country and that they will continue to spread along with the growing cases. To support the latter, as of 22 April 2020 we downloaded 1,599 GenBank sequences of SARS-CoV-2 from Asia, Europe and North America origin, respectively and we aligned them against the SARS-CoV-2 reference genome. A phylogenetic tree was constructed from all genbank sequences and depict a mixed clustering of sequences between Asia, Europe and North America, supporting the existance of different viral signatures. (Fig. S1). Variant calling from these alignment reveals the substantial presence of USA-WA signature in North America sequences, with allele frequencies (AF) ranging 33–39% (see Fig. 1E; Table S1, sheet 6). Variants 8,782 and 28,144 from USA-WA signature are also present in Asia and Europe, suggesting these two variants arises in the beginning of the pandemic and spread worldwide. Thus, the USA-WA signature is likely widespread among SARS-CoV-2 infections in USA due to founder effect. Also, in North America GenBank sequences, six more variants were detected with similar allele frequencies as reported for USA-WA variants. A summary of the USA-WA mutational signature is depicted in Table 1. Less founder variants with lower allele frequencies are present in Asian samples suggesting high clonality of the original strain of SARS-CoV-2. Conversely, in Europe 16 founder variants with higher allele frequencies were found, supporting SARS-CoV-2 evolves as the pandemic spread. Overall, missense GenBank variants equals or surpasses synonymous variants in ORF1ab,

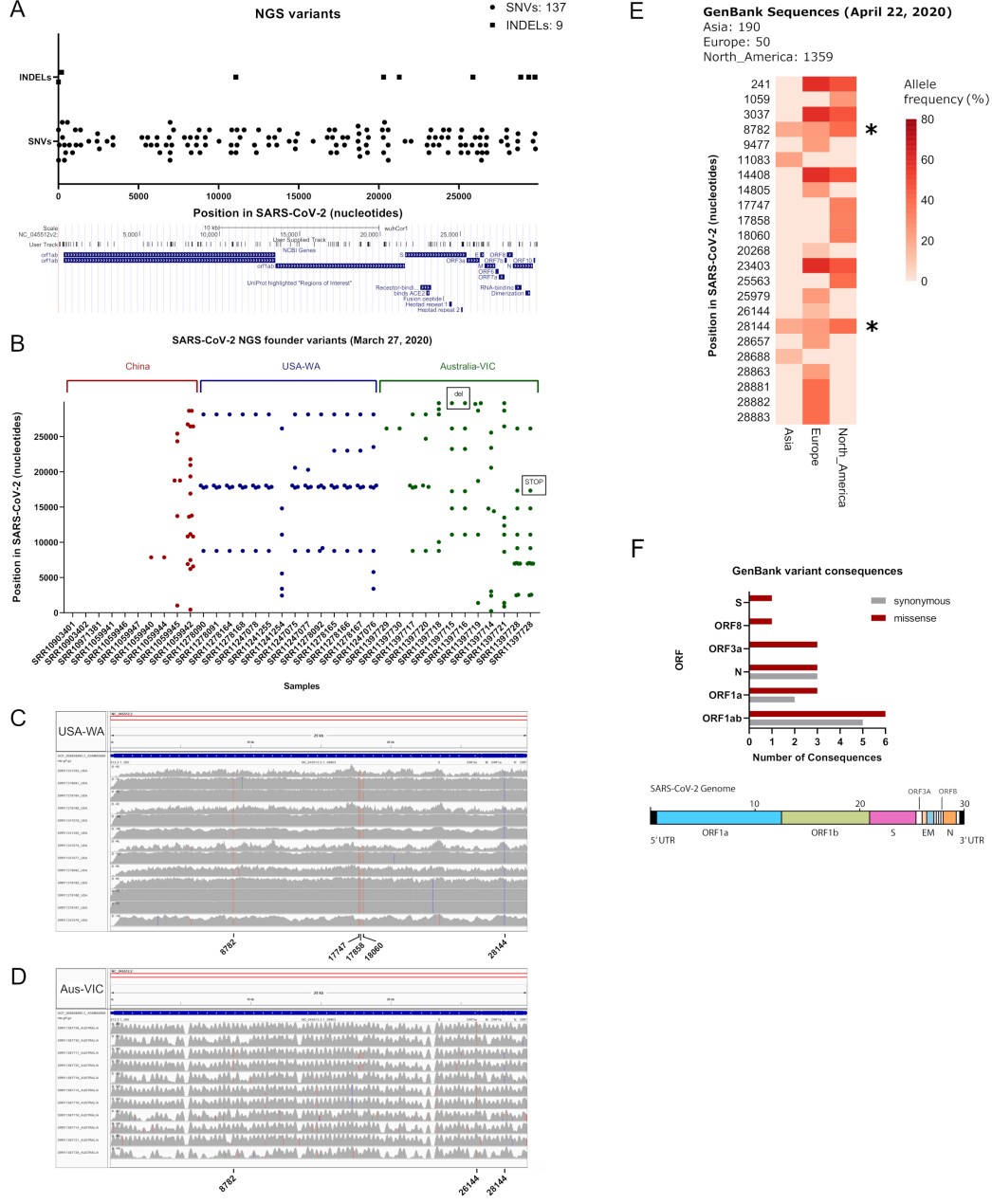

**Figure 1  Inspection of variants reveals well-defined signatures with founder effect across sequenced samples.** (A) (Upper) Plot of all merged variants from NGS datasets ($n = 50$) depicting single nucleotide variants (black dots) and indels (black squares) along SARS-CoV-2 nucleotide positions. (Lower) Snapshot of SARS-CoV-2 ORFs and receptor binding domains. User track denotes merged NGS variants across ORFs. (B) Plot of founder variants sorted by country (China = red, USA = blue and Australia = green) and by the number of variants, from left to right. Deletions and stop codons are framed with black rectangles. (C) IGV screenshots of coverage from USA samples ($n = 13$) aligned against SARS-CoV-2 reference genome. Founder variants are depicted with colored lines. (D) IGV screenshots of coverage from Australian samples ($n = 11$) aligned against SARS-CoV-2 reference genome. Founder variants are depicted with colored lines. (E) Allele frequency (plotted as percentage) of founder variants collected from GenBank SARS-CoV-2 sequence alignments from Asia ($n = 270$), Europe ($n = 50$) and North America ($n = 1,359$). Asterisks denotes common founder variants in the three regions, part of the USA-WA signature (8,782, 28,144). (F) (Upper) Variant consequence classification of GenBank
**Figure 1** (continued)
founder variants obtained with the variant annotator integrator tool. Missense and synonymous variant consequences per ORF are denoted with red grey bars, respectively. (Lower) Schematic diagram of the general genetic composition of SARS-CoV-2. Colored boxes correspond to main genes and white boxes to smaller ORFs.                                     

**Table 1 Allele frequencies of five detected variants in 1,359 GenBank sequences from North America.**

| POS | REF | ALT | INFO | AF |
|---|---|---|---|---|
| 8782 | C | T | DP = 302; DP4 = 246,0,161,0 | 39.55774 |
| 17747 | C | T | DP = 302; DP4 = 272,0,135,0 | 33.16953 |
| 17858 | A | G | DP = 302; DP4 = 273,0,134,0 | 32.92383 |
| 18060 | C | T | DP = 302; DP4 = 271,0,136,0 | 33.41523 |
| 28144 | T | C | DP = 303; DP4 = 246,0,161,0 | 39.55773 |

Note:
 POS, position in SARS CoV 2 reference sequence NC_045512.2; REF, reference allele; ALT, mutant allele; DP4, Number of high quality ref forward, ref reverse, alt forward and alt reverse bases; AF, allele frequency (%).

Nucleocapsid (N), ORF3a, ORF8 and surface glycoprotein (S) (see Fig. 1F; Table S1, sheet 7). Thus, as the pandemic evolves, successful molecular adaptations in SARS-CoV-2 also occurred as is presented in the different viral signatures.

## Classification of next-generation sequencing variants

To see if the latter observations are replicated in next generation sequencing datasets, we aimed to characterize in depth USA-WA and Australia-VIC variants as performed with founder variants from GenBank sequences. In agreement with the latter, classification of variants performed by snippy tool analysis reveals that most variants in USA-WA and Australia-VIC are preferentially missense (63% for USA-WA samples and 74% for Australia-VIC samples, respectively) rather than synonymous (see Figs. 2A and 2B; Table S1, sheets 4 and 5, respectively). Focusing on missense variants of USA-WA samples, the most recurrent mutations occurred in the polypeptide ORF1ab (encoding a viral helicase and 3'–5' exonuclease) and ORF8 protein, accounting for the 81% of missense variants in the USA-WA signature (see Fig. 2C; Table S1, sheet 4). In the case of Australia-VIC missense variants, the scenario is more complex, due to the heterogeneity of the signatures. Mutations in the ORF1a polyprotein, ORF3a protein, helicase and surface glycoprotein account for 69% of the missense variants present in the Australia-VIC signatures (see Fig. 2D). Importantly, two Australian samples (SRR11397715 and SRR11397716) present the same mutational profiling with deletions in the stem loop of the virus and notably, one sample (SRR11397728) presents a SNP that creates a stop codon in the helicase protein (see Table S1, sheet 5). Since every USA-WA and Australia-VIC sample that presents variants in the helicase gene contains at least one missense variant, this evidence strongly suggests that this gene is located in an actively evolving region from SARS-CoV-2 and it will likely continue to evolve as the pandemic spreads.

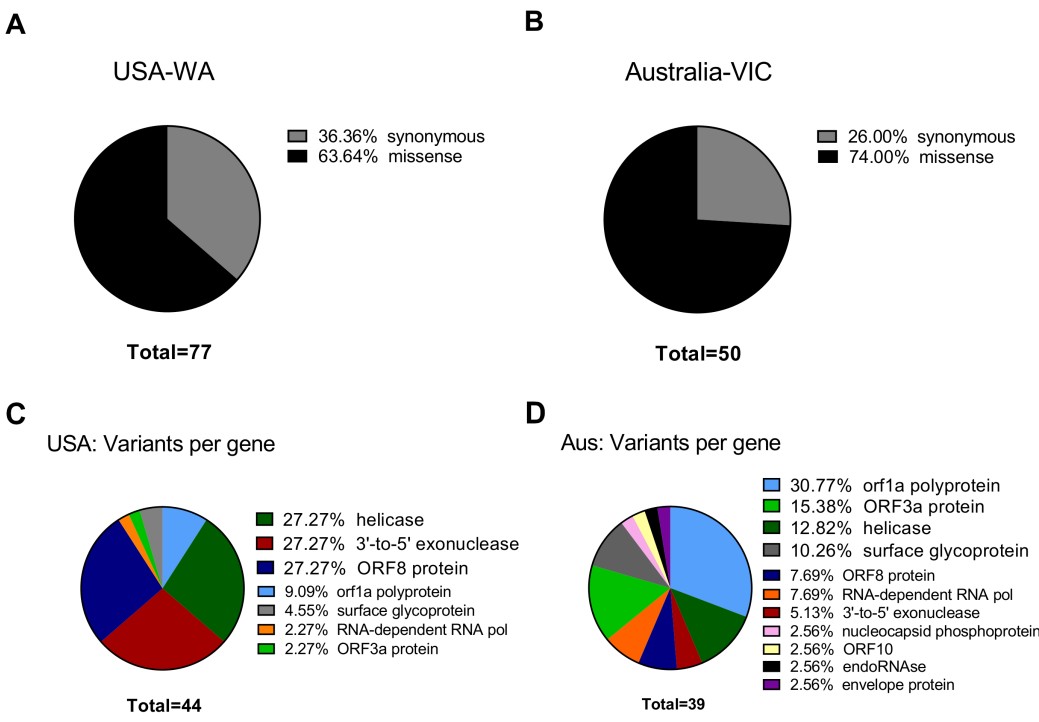

**Figure 2 Classification of SARS-CoV-2 variants from next-generation sequencing datasets.** (A) Ratio of Synonymous/missense variants across USA samples. Missense variants are denoted in black and Synonymous variants are depicted in grey. N indicates the total number of classified variants. (B) Same as left for Australian samples. (C) Missense variants classification from USA samples across SARS-CoV-2 genes. Each color indicates a different gene stated in legend (at right of the pie charts). (D) Same as left from Australian samples.

## SNPs in SARS-CoV-2 may diminish efficiency of RT-qPCR testing

The Centers for Disease Control and Prevention primer list, consists of three primer sets designed against the ORF9 structural protein (nucleocapsid phosphoprotein), each one with a fluorescent probe for reverse transcriptase quantitative PCR. We investigated if these primers hybridized at positions that fall within the variants reported herein from the 50 next-generation sequencing datasets. We aligned these primer sequences with SARS-CoV-2 reference genome and the 50 analyzed samples. We found two Australian clonal samples (SRR11397719 and SRR11397721) presenting one founder synonymous SNP (position 28,688) that occurs within the 2019_nCoV_N3_Forward_Primer hybridization region. In the first sample, we also found an SNP with low allele frequency that falls into the 2019-nCoV_N1_Reverse_Primer (see Fig. 3). We also challenged a list of primers available from literature, primers currently used for the Institute Pasteur, Hong Kong University and ten primers obtained from primer-blast against merged GenBank variants, including variants from the next-generation sequencing datasets employed here. Of these, CDC primer set 2019-nCoV_N3 forward and reverse along with its probe were discarded again, due to potential reduced efficiency during priming (see Table S1, sheet 8). Similarly, sets 1, 2, 3 and 8 from primer-blast, were discarded. Conversely, primers used by the Pasteur Institute and Hong-Kong university passed the

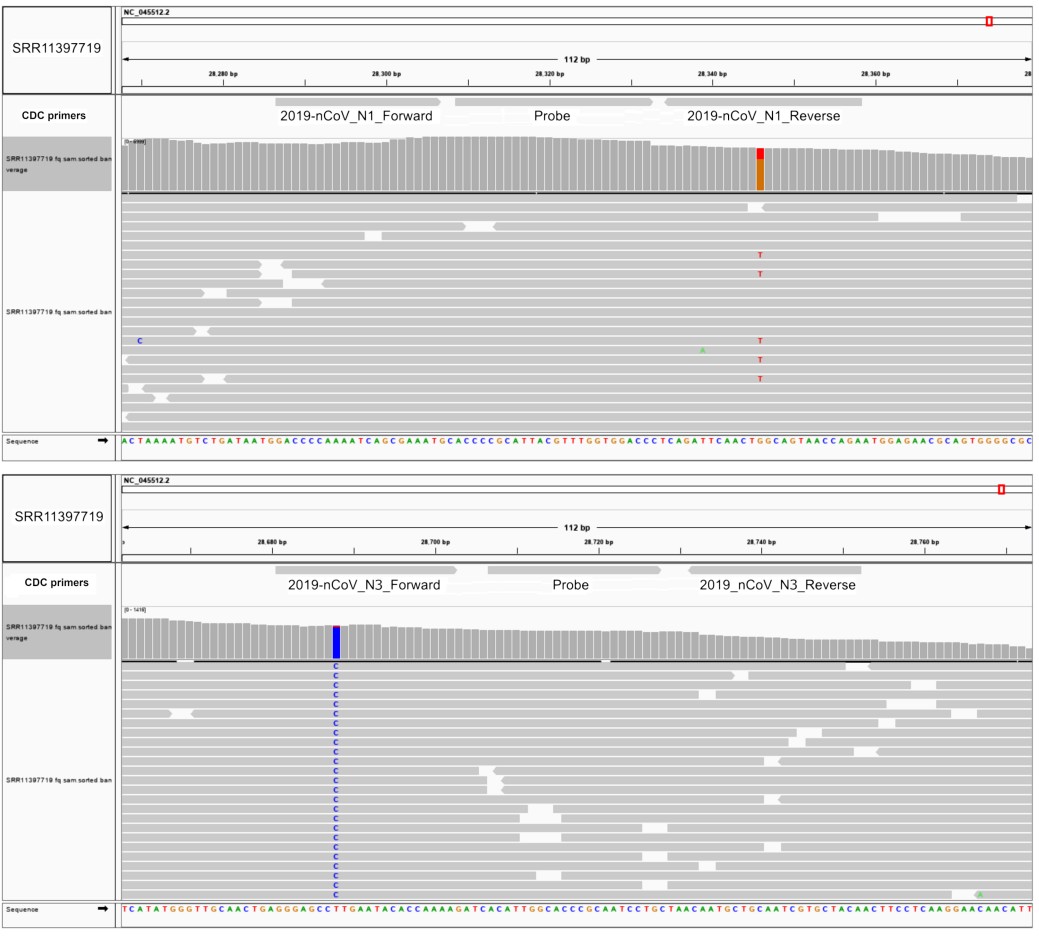

**Figure 3 SNPs in SARS-CoV-2 may diminish efficiency of RT-qPCR testing.** Top IGV screenshots of coverage from Australian sample SRR11397719 aligned against SARS-CoV-2 reference genome. A low allele frequency variant is depicted in red (T). Primers tracks are denoted at the top of the screenshot along with SARS-CoV-2 gff gene models. Bottom same as top for sample SRR11397719 denoting a founder variant in blue.

filter, respectively. Thus, increasing variation in SARS-CoV-2 can confound quantitative RT-qPCR in the future depending on the primer design.

## DISCUSSION

In this study, we have analyzed early mutational events occurring in SARS-CoV-2 illumina whole-genome sequencing samples from different populations (USA, Australia and China) and we compared these results with variants observed in submitted GenBank sequences in NCBI viral portal up until 22 April 2020. As already reported with HIV and Chikungunya outbreaks, the founder effect of five-point variants was observed in almost all USA-WA samples obtained by next-generation sequencing. These mutations also have high allele frequencies (around 33–39%) in SARS-CoV-2 GenBank sequences from USA origin (*Foley et al., 2000*; *Tsetsarkin et al., 2011*). These variant signatures are likely to be overrepresented among Washington State infections and USA infections overall, if not globally. Supporting the latter, the alignment of 1359 SARS-CoV-2 submitted

sequences to NCBI virus from North America (22 April 2020) shows allele frequencies of 33–39% of these five SNPs across samples worldwide. These SNPs cause missense mutations in helicase, 3′–5′ exonuclease and ORF8 proteins. In the case of Victoria samples from Australia, founder variants from one up to eleven SNPs were found. In the early beginning of the outbreak in Wuhan city (between 18 and 29 December 2019), one to four mutations arose in the virus per patient (*Shen et al., 2020*), arguing that the number of fixed mutations in the world population is rapidly increasing where the infection has spread. Importantly, one Australian sample sequenced by next generation sequencing presented the USA-WA signature, suggesting that this signature is already propagated with the worldwide pandemic. Consistent with the latter, real-time tracking of pathogen evolution and phylogenetic analysis provided from the Nextstrain initiative demonstrated dissemination of SARS-CoV-2 viral signatures from USA to Australia and Europe in early February 2020 (*Hadfield et al., 2018*). Clonal mutational events within Australian samples were also observed and probably are widespread in the region. One interesting feature from USA-WA signature are missense variants occurring in the helicase gene (ORF1ab). RNA helicases display various functions in genome replication, they have even been proposed as a therapeutic target to inhibit coronaviruses among other viruses with small molecules (*Briguglio et al., 2011*), thus mutations evoked by these variants could make drug targeting this protein more difficult in the future. Also, as genetic drift is allowing SARS-CoV-2 to evolve as the pandemic continues, the amplification efficiency of quantitative RT-qPCR tests may be compromised since a single mutational even in the middle of a primer sequence can be detrimental for PCR efficiency (*Bru, Martin-Laurent & Philippot, 2008*), potentially contributing to false negative results in COVID-19 testing. For this issue, we provided a way to compute the latter, by merging all variant sites called across studied samples and by intersecting them across primer sets available both in the literature and currently in use in viral testing kits. As new mutations can be spread depending on the founder effect, we firmly urge that primer sets for clinical testing should be tested in this way continuously, according to the current mutations found at the particular time and in the specific population which needs to be diagnosed with SARS-CoV-2 infection.

## CONCLUSIONS

We describe here the early mutational events in SARS-CoV-2 virus by analyzing sequencing samples from China, USA, Australia and GenBank sequences submitted between 27 March and 22 April 2020. SARS-CoV-2 variants from the USA display five-point mutations with clonal (founder) patterns of spreading at a considerably high frequency among samples. The latter was verified by sequence analysis of SARS-CoV-2 sequences submitted to GenBank, since these five-point mutations displayed alleles frequencies of 33–39% among all USA GenBank SARS-CoV-2 sequences ($n = 1,359$). SARS-CoV-2 Australian variants were heterogeneous, but still, clonal events were found including one sample presenting the USA-WA signature, implying worldwide spreading of this signature. The efficiency of RT-qPCR testing can be potentially affected by founder variants, since several SNPs affecting one of three primers sets currently used in

COVID-19 testing has been found. By the time of this publication, the available data could change the conclusions presented in this manuscript as a result of further viral variants arising.

## ACKNOWLEDGEMENTS

We thank 'Convenio de colaboración diagnóstico y otras actividades relativas al COVID-19', Agencia Nacional de Investigacion y Desarrollo (ANID), PROYECTO COVID ID18I10261.

### Funding

The Canadian Institute of Health Research (CIHR, project grant 419220) funded this work. The funders had no role in study design, data collection and analysis, decision to publish, or preparation of the manuscript.

### Grant Disclosures

The following grant information was disclosed by the authors:
Canadian Institute of Health Research (CIHR): 419220.

### Competing Interests

José Luis Garrido is a co-founder of Ichor Biologics LLC and is currently employed by Ichor Biologics LLC, New York, United States.

The rest of the authors currently work in the academia and do not belong to industrial/comercial enterprises. All of the authors declared that they have no competing interests.

### Author Contributions

- Carlos Farkas conceived and designed the experiments, performed the experiments, analyzed the data, prepared figures and/or tables, authored or reviewed drafts of the paper, and approved the final draft.
- Francisco Fuentes-Villalobos analyzed the data, authored or reviewed drafts of the paper, and approved the final draft.
- Jose Luis Garrido analyzed the data, authored or reviewed drafts of the paper, and approved the final draft.
- Jody Haigh conceived and designed the experiments, analyzed the data, authored or reviewed drafts of the paper, and approved the final draft.
- María Inés Barría conceived and designed the experiments, analyzed the data, authored or reviewed drafts of the paper, and approved the final draft.

### Data Availability

All next generation sequencing dataset accessions and resulting computational calculations are available in Table S1. All commands to obtain these computational steps

are publicly available at GitHub: https://github.com/cfarkas/SARS-CoV-2_illumina_analysis. The repository can be downloaded freely.

## Supplemental Information

Supplemental information for this article can be found online at http://dx.doi.org/10.7717/peerj.9255#supplemental-information.

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
