# Peer review of "Insights on early mutational events in SARS-CoV-2 virus reveal founder effects across geographical regions"

_PeerJ, doi:10.7717/peerj.9255_

## Round 0.1 · original submission · Minor Revisions

The reviews are heterogeneous but on balance they are mostly favorable, except for reviewer 3. I would encourage you to revise your paper quickly so that it can be published, otherwise it will quickly become outdated. You should probably include a statement that by the time of publication, the available data could change the conclusions presented.

Reviewer 1 ·

Basic reporting

The ‘Basic Reporting’ features from PeerJ are generally well met. The writing does includes more than the usual number of typos or poorly constructed sentences. For example, the sentence starting on line 262 needs revision (“One interesting feature of the mutations in this limited number of samples is the evolutionary pressure undergone helicase gene.”). Also see line 266 (“…making it unfeasible to drug this protein…”)

Experimental design

The experimental design features from PeerJ appear to be met.

Validity of the findings

The validity of findings features from PeerJ also appear to be met.

Additional comments

The submission by Farkas et al. (Insights on early mutational events…) is a straightforward report on the point substitutions found in an early set of SARS-CoV-2 virus genomes downloaded from public databases. The main findings are that, within their sampling scheme, (1) a set of point mutations, with high allelic frequencies, distinguish viruses from Washington state in the US relative to the first genomes from China, (2) a set of Australian virus genomes also showed clonal mutations but these were more heterogeneous, and (3) some of the point substitutions occurred in genome locations targeted by PCR primers. They warn that PCR primers will need continual evaluation to provide reliable tests for infection.
Line 63 states, “…it is likely that pangolin species are a natural reservoir…” That is not my reading of the recent studies. I’d refer the authors to Zhang and Holmes 2020. Cell; https://doi.org/10.1016/j.cell.2020.03.035 and references therein, which suggest that pangolins are better characterized as a possible intermediate host in transmission of coronaviruses between bats and humans. Bats are better characterized as natural hosts for coronaviruses. Pangolins populations (endangered) have not been extensively evaluated for coronaviruses.
The interpretation of the findings are limited and reasonable, though I don’t think the existence of point mutations alone constitutes “evolutionary pressure” (line 263), which usually connotes directional selection. If using the term the authors would need to show that the changes are not random in their distribution and overall population effects.

Reviewer 2 ·

Basic reporting

The Basic reporting of the study is correct but please see "General comments for the author".

Experimental design

The Experimental design of the study is correct but please see "General comments for the author".

Validity of the findings

The Validity of the findings of the study is correct but please see "General comments for the author".

Additional comments

This study investigates the presence of founder effects in the current expansion of the SARS-CoV-2 virus using on early mutational events. The study is interesting but I find several aspects that should be clarified. My recommendation is accept after revisions.

Major comments

I am wondering about the nucleotide diversity of the multiple sequence alignment of the 436 genome sequences downloaded from the NCBI. My impression (but just an impression) is that the nucleotide diversity is low, and in this case it could affect the detection of founder effects because lack of genetic information.

I am wondering if the results could change with the consideration of more genome sequences in the studied dataset, particularly sequences from other locations (i.e., Europe).

Minor comments

In some places along the manuscript it is mentioned that this virus is rapidly evolving worldwide. However it is unclear if this affirmation is based on the number of infections, number of mutations or number of fixed mutations (substitutions), etc. Is there any estimation about the genetic evolutionary rate of this virus to affirm that this is rapidly evolving virus variant?. If not I suggest modify the text.

In some places along the manuscript a sentence like the following is “Mutations in the helicase and orf1a coding regions from SARS-CoV-2 were predominant, among others, suggesting that these proteins are prone to evolve by natural selection”. I do not see the relationship between “the number of mutations” and “natural selection”. Natural selection can occur at any genetic region based on molecular adaptation (i.e., evolutionary constraints from the function), not on the number of mutations. Slowly evolving genetic regions are also subjected to the action of natural selection. I suggest modify the text.

The “Results” section repeats some information already presented in the “Methods” section. It needs to be rewritten avoiding repetitions.

Since the text mentions specific protein-coding regions (i.e., helicase, pol, etc), it would be interesting for readers to add a figure (either for the main text or for the supplementary material) illustrating the different regions of this virus, just something like, https://en.wikipedia.org/wiki/Structure_and_genome_of_HIV#/media/File:HIV-genome.png and highlighting the regions mentioned in this study. This could help readers to find the location of the studied regions in the whole genome. This is a suggestion.

It could be nice for readers if they can download the analyzed multiple sequence alignment (436 genome sequences downloaded from the NCBI) directly, from a link given by the authors (i.e., to GitHub or any other datasets repository). Indeed, note that the journal asks us (reviewers) about the raw data used in the study.

Reviewer 3 ·

Basic reporting

There is not enough justification for using the selected samples for analyses in order to confirm/refute that the identified mutations represent "founder" effects in the corresponding countries.

Experimental design

The experimenal design is flawed and cannot be used to test the hypotheses posed by the authors.

Validity of the findings

The conclusions are not sustained by the data presented because these have not been analyzed adequately.

Additional comments

This manuscript presents an analysis of a very limited number of SARS-CoV-2 isolates from China, USA and Australia selected from the availability of sequencing projects deposited at the NCBI SRA repository. Raw reads from 30 samples were used to identify variants against the reference used for mapping (one of the reference genomes initially obtained in Wuhan). Then, the frequency of these variants was estimated by downloading sequences from the corresponding countries deposited in GenBank. The analysis proceeds with a detailed characterization of each mutation as inferred from the annotation of the corresponding loci and its effects in the deduced protein sequence when in a CDS.
In my opinion, this study has several major flaws that lead me to recommend its rejection. Firstly, there is no clear definition of what the authors understand by “founder mutation” and how this is established in the corresponding populations. From their logic, it seems that founder mutations are those found in the 30 samples finally used for analysis, with no justification for why these and not other samples are the ones that seed the corresponding clades in each country. In addition, there is a no provision for population genetic nor phylogenetic analyses of the fate of each of these mutations when compared with the extant variability inferred for samples for each of the clades where the samples are included. Using a few samples from a bioproject (all of which have only a few samples) to establish any “founder effect” lacks logic and justification.
There are several additional issues on how to analyze population data but lack of information on the phylogenetic and epidemiological relationships among the viral sequences used in the analyses makes the conclusions of this manuscript completely unjustified.

---

## Round 0.2 · Minor Revisions

Please make some reference to the ongoing Nextstrain coronavirus analysis available online. https://nextstrain.org/ Does this resource place your results in context?

Also, please define in the main text what you mean by "germline variants". You use this term in the legend to Fig. 1 but you don't define it and it's not clear what it means.

Once these revisions are in place it will be acceptable for publication.

---

## Round 0.3 · accepted · Accept

Thank you for addressing the comments.